# GENERALIZED MULTIMODAL ELBO

**Thomas M. Sutter**[*]     **Imant Daunhawer**[*]     **Julia E. Vogt**

Department of Computer Science
ETH Zurich
8092 Zurich, Switzerland
{thomas.sutter,imant.daunhawer,julia.vogt}@inf.ethz.ch

## ABSTRACT

Multiple data types naturally co-occur when describing real-world phenomena and learning from them is a long-standing goal in machine learning research. However, existing self-supervised generative models approximating an ELBO are not able to fulfill all desired requirements of multimodal models: their posterior approximation functions lead to a trade-off between the semantic coherence and the ability to learn the joint data distribution. We propose a new, generalized ELBO formulation for multimodal data that overcomes these limitations. The new objective encompasses two previous methods as special cases and combines their benefits without compromises. In extensive experiments, we demonstrate the advantage of the proposed method compared to state-of-the-art models in self-supervised, generative learning tasks.

## 1 INTRODUCTION

The availability of multiple data types provides a rich source of information and holds promise for learning representations that generalize well across multiple modalities (Baltrušaitis et al., 2018). Multimodal data naturally grants additional self-supervision in the form of shared information connecting the different data types. Further, the understanding of different modalities and the interplay between data types are non-trivial research questions and long-standing goals in machine learning research. While fully-supervised approaches have been applied successfully (Karpathy & Fei-Fei, 2015; Tsai et al., 2019; Pham et al., 2019; Schoenauer-Sebag et al., 2019), the labeling of multiple data types remains time consuming and expensive. Therefore, it requires models that efficiently learn from multiple data types in a self-supervised fashion.

Self-supervised, generative models are suitable for learning the joint distribution of multiple data types without supervision. We focus on VAEs (Kingma & Welling, 2014; Rezende et al., 2014) which are able to jointly infer representations and generate new observations. Despite their success on unimodal datasets, there are additional challenges associated with multimodal data (Suzuki et al., 2016; Vedantam et al., 2018). In particular, multimodal generative models need to represent both modality-specific and shared factors and generate semantically coherent samples across modalities. Semantically coherent samples are connected by the information which is shared between data types (Shi et al., 2019). These requirements are not inherent to the objective—the evidence lower bound (ELBO)—of unimodal VAEs. Hence, adaptions to the original formulation are required to cater to and benefit from multiple data types. Furthermore, to handle missing modalities, there is a scalability issue in terms of the number of modalities: naively, it requires $2^M$ different encoders to handle all combinations for $M$ data types. Thus, we restrict our search for an improved multimodal ELBO to the class of *scalable* multimodal VAEs.

Among the class of scalable multimodal VAEs, there are two dominant strains of models, based on either the multimodal variational autoencoder (MVAE, Wu & Goodman, 2018) or the Mixture-of-Experts multimodal variational autoencoder (MMVAE, Shi et al., 2019). However, we show that these approaches differ merely in their choice of joint posterior approximation functions. We draw a theoretical connection between these models, showing that they can be subsumed under the class

---

[*]Equal contribution.

of abstract mean functions for modeling the joint posterior. This insight has practical implications, because the choice of mean function directly influences the properties of a model (Nielsen, 2019). The MVAE uses a geometric mean, which enables learning a sharp posterior, resulting in a good approximation of the joint distribution. On the other hand, the MMVAE applies an arithmetic mean which allows better learning of the unimodal and pairwise conditional distributions. We generalize these approaches and introduce the Mixture-of-Products-of-Experts-VAE that combines the benefits of both methods without considerable trade-offs.

In summary, we derive a generalized multimodal ELBO formulation that connects and generalizes two previous approaches. The proposed method, termed MoPoE-VAE, models the joint posterior approximation as a Mixture-of-Products-of-Experts, which encompasses the MVAE (Product-of-Experts) and MMVAE (Mixture-of-Experts) as special cases (Section 3). In contrast to previous models, the proposed model approximates the joint posterior for *all subsets* of modalities, an advantage that we validate empirically in Section 4, where our model achieves state-of-the-art results.

## 2 RELATED WORK

This work extends and generalizes existing work in self-supervised multimodal generative models that are scalable in the number of modalities. Scalable in the sense that a single model approximates the joint distribution over all modalities (including all marginal and conditional distributions) instead of requiring individual models for every subset of modalities (e.g., Huang et al., 2018; Tian & Engel, 2019; Hsu & Glass, 2018). The latter approach requires a prohibitive number of models, exponential in number of modalities.

**Multimodal VAEs**  Among multimodal generative models, multimodal VAEs (Suzuki et al., 2016; Vedantam et al., 2018; Kurle et al., 2019; Tsai et al., 2019; Wu & Goodman, 2018; Shi et al., 2019; 2020; Sutter et al., 2020) have recently been the dominant approach. Multimodal VAEs are not only suitable to learn a joint distribution over multiple modalities, but also enable joint inference given a subset of modalities. However, to approximate the joint posterior for all subsets of modalities efficiently, it is required to introduce additional assumptions on the form of the joint posterior. To overcome the issue of scalability, previous work relies on either the product (Kurle et al., 2019; Wu & Goodman, 2018) or the mixture (Shi et al., 2019; 2020) of unimodal posteriors. While both approaches have their merits, there are also disadvantages associated with them. We unite these approaches in a generalized formulation—a mixture of products joint posterior—that encapsulates both approaches and combines their benefits without significant trade-offs.

**Multimodal posteriors**  The MVAE (Wu & Goodman, 2018) assumes that the joint posterior is a product of unimodal posteriors—a Product-of-Experts (PoE, Hinton, 2002). The PoE has the benefit of aggregating information across any subset of unimodal posteriors and therefore provides an efficient way of dealing with missing modalities for specific types of unimodal posteriors (e.g., Gaussians). However, to handle missing modalities the MVAE relies on an additional sub-sampling of unimodal log-likelihoods, which no longer guarantees a valid lower bound on the joint log-likelihood (Wu & Goodman, 2019). Previous work provides empirical results that exhibit the shortcomings of the MVAE, attributing them to a precision miscalibration of experts (Shi et al., 2019) or to the averaging over inseparable individual beliefs (Kurle et al., 2019). Our results suggest that the PoE works well in practice, if it is also applied on all subsets of modalities, which naturally leads to the proposed Mixture-of-Products-of-Experts (MoPoE) generalization, which yields a valid lower bound on the joint log-likelihood.

On the other hand, the MMVAE (Shi et al., 2019) assumes that the joint posterior is a mixture of unimodal posteriors—a Mixture-of-Experts (MoE). The MMVAE is suitable for the approximation of unimodal posteriors and for translation between pairs of modalities, however, it cannot take advantage of multiple modalities being present, because it only takes the unimodal posteriors into account during training. In contrast, the proposed MoPoE-VAE computes the joint posterior for all subsets of modalities and therefore enables efficient *many-to-many* translations. Extensions of the MVAE and MMVAE (Kurle et al., 2019; Daunhawer et al., 2020; Shi et al., 2020; Sutter et al., 2020) have introduced additional loss terms, however, these are also applicable to and can be added on top of the proposed model.

Table 1: Properties of previous scalable multimodal VAEs and our proposed model. Note that to deal with missing modalities, the MVAE requires sub-sampling of unimodal ELBOs, which yields an invalid bound on the joint log-likelihood (Wu & Goodman, 2019).

| Model | Posterior form | Aggregate modalities | Multi-modal posterior | Missing modalities |
|-------|----------------|----------------------|-----------------------|--------------------|
| MVAE | PoE | ✓ | ✗ | (✓) |
| MMVAE | MoE | ✗ | ✓ | ✓ |
| MoPoE-VAE (ours) | MoPoE | ✓ | ✓ | ✓ |

Table 1 summarizes the properties of previous multimodal VAEs and highlights the benefits of the proposed model: the ability to aggregate multiple modalities, to learn a multi-modal posterior (in the statistical sense), and to efficiently handle missing modalities at test time.

## 3 METHOD

### 3.1 PRELIMINARIES

We consider a dataset $\{\mathbb{X}^{(i)}\}_{i=1}^N$ of $N$ i.i.d. samples, each of which is a set of $M$ modalities $\mathbb{X}^{(i)} = \{\boldsymbol{x}_j^{(i)}\}_{j=1}^M$. We assume that the data is generated by some random process involving a joint hidden random variable $\boldsymbol{z}$ such that inter-modality dependencies are unknown. The marginal log-likelihood can be decomposed into a sum over marginal log-likelihoods of individual sets $\log p_\theta(\{\mathbb{X}^{(i)}\}_{i=1}^N) = \sum_{i=1}^N \log p_\theta(\mathbb{X}^{(i)})$, which can be written as:

$$\log p_\theta(\mathbb{X}^{(i)}) = D_{\mathrm{KL}}(q_\phi(\boldsymbol{z}|\mathbb{X}^{(i)})||p_\theta(\boldsymbol{z}|\mathbb{X}^{(i)})) + \mathcal{L}(\theta, \phi; \mathbb{X}^{(i)}), \tag{1}$$

$$\text{with } \mathcal{L}(\theta, \phi; \mathbb{X}^{(i)}) := \mathbb{E}_{q_\phi(\boldsymbol{z}|\mathbb{X}^{(i)})}[\log p_\theta(\mathbb{X}^{(i)}|\boldsymbol{z})] - D_{\mathrm{KL}}(q_\phi(\boldsymbol{z}|\mathbb{X}^{(i)})||p_\theta(\boldsymbol{z})). \tag{2}$$

$\mathcal{L}(\theta, \phi; \mathbb{X}^{(i)})$ is called evidence lower bound (ELBO) on the marginal log-likelihood of the $i$-th set. It forms a tractable objective to approximate the joint data distribution $\log p_\theta(\mathbb{X}^{(i)})$. $q_\phi(\boldsymbol{z}|\mathbb{X}^{(i)})$ is the posterior approximation distribution with learnable parameters $\phi$. From the non-negativity of the KL divergence, it follows that $\log p_\theta(\mathbb{X}^{(i)}) \geq \mathcal{L}(\theta, \phi; \mathbb{X}^{(i)})$. If the posterior approximation $q_\phi(\boldsymbol{z}|\mathbb{X}^{(i)})$ is identical to the true posterior distribution $p_\theta(\boldsymbol{z}|\mathbb{X}^{(i)})$, the bound holds with equality. Hence, maximizing the ELBO in Equation (2) minimizes the otherwise intractable KL-divergence between approximate and true posterior distribution:

$$\arg\min_\phi D_{\mathrm{KL}}(q_\phi(\boldsymbol{z}|\mathbb{X}^{(i)})||p_\theta(\boldsymbol{z}|\mathbb{X}^{(i)})). \tag{3}$$

Adaptations to the ELBO formulation in Equation (2) include an additional hyperparameter $\beta$ which weights the KL-divergence relative to the log-likelihood (Higgins et al., 2017). To improve readability, we will omit the superscript $(i)$ in the remaining part of this work.

### 3.2 APPROXIMATING $p_\theta(z|\mathbb{X})$ IN CASE OF MISSING DATA TYPES

For a dataset of $M$ modalities, there are $2^M$ different subsets contained in the powerset $\mathcal{P}(\mathbb{X})$. If, for a particular observation, we only have access to a subset of data types $\mathbb{X}_k \in \mathcal{P}(\mathbb{X})$, the approximation of $p_\theta(\mathbb{X}_k)$ would result in a different ELBO formulation $\mathcal{L}(\theta, \phi_k; \mathbb{X}_k)$ where the true posterior $p_\theta(\boldsymbol{z}|\mathbb{X}_k)$ of subset $\mathbb{X}_k$ is approximated. Instead, we are interested in the true posterior $p_\theta(\boldsymbol{z}|\mathbb{X})$ of all data types $\mathbb{X}$, even when only a subset $\mathbb{X}_k$, i.e. $\tilde{q}_{\phi_k}(\boldsymbol{z}|\mathbb{X}_k)$, is available. The desired ELBO for the available subset $\mathbb{X}_k$ is given by

$$\mathcal{L}_k(\theta, \phi_k; \mathbb{X}) := \mathbb{E}_{\tilde{q}_{\phi_k}(\boldsymbol{z}|\mathbb{X}_k)}[\log(p_\theta(\mathbb{X}|\boldsymbol{z})] - D_{\mathrm{KL}}(\tilde{q}_{\phi_k}(\boldsymbol{z}|\mathbb{X}_k)||p_\theta(\boldsymbol{z})). \tag{4}$$

The subtle but important difference between $\mathcal{L}_k(\theta, \phi_k; \mathbb{X})$ and $\mathcal{L}(\theta, \phi_k; \mathbb{X}_k)$ is that the former still yields a valid lower bound on $p_\theta(\mathbb{X})$, whereas the latter forms a lower bound on $\log p_\theta(\mathbb{X}_k)$, which is no longer a valid bound on the desired $\log p_\theta(\mathbb{X})$.

Different from previous work, we argue for an optimization of the powerset $\mathcal{P}(\mathbb{X})$, i.e., the joint optimization of all ELBOs $\mathcal{L}_k(\theta, \phi_k; \mathbb{X})$ defined by the posterior subset approximation $\tilde{q}_{\phi_k}(\boldsymbol{z}|\mathbb{X}_k)$. Since maximizing the ELBO in Equation (2) is equivalent to minimizing the KL-divergence in Equation (3), the joint optimization of the powerset $\mathcal{P}(\mathbb{X})$ is equal to the minimization of the following convex combination of KL-divergences of the power set $\mathcal{P}(\mathbb{X})$.[1]

$$\arg\min_{\phi} \sum\nolimits_{\mathbb{X}_k \in \mathcal{P}(\mathbb{X})} D_{\mathrm{KL}}(\tilde{q}_\phi(\boldsymbol{z}|\mathbb{X}_k)||p_\theta(\boldsymbol{z}|\mathbb{X})) \tag{5}$$

Hence, we propose to optimize Equation (4) for all subsets $\mathbb{X}_k$.

**Lemma 1.** *The sum of KL-divergences in Equation* (5) *describes the joint probability* $\log p_\theta(\mathbb{X})$ *as follows:*

$$\log p_\theta(\mathbb{X}) = \frac{1}{2^M} \sum_{\mathbb{X}_k \in \mathcal{P}(\mathbb{X})} D_{\mathrm{KL}}\left(\tilde{q}_\phi(\boldsymbol{z}|\mathbb{X}_k)||p_\theta(\boldsymbol{z}|\mathbb{X})\right) + \frac{1}{2^M} \sum_{\mathbb{X}_k \in \mathcal{P}(\mathbb{X})} \mathbb{E}_{\tilde{q}_\phi(\boldsymbol{z}|\mathbb{X}_k)} \left[ \log \frac{p_\theta(\mathbb{X}|\boldsymbol{z})p_\theta(\boldsymbol{z})}{\tilde{q}_\phi(\boldsymbol{z}|\mathbb{X}_k)} \right]$$

Following Lemma 1 (see Appendix A.1 for the proof) and the non-negativity of the KL-divergence, we see that the convex combination of expectations over the powerset $\mathcal{P}(\mathbb{X})$ is an ELBO on the joint probability $\log p_\theta(\mathbb{X})$. Since this would require $2^M$ different inference networks in a naive implementation, we use a more efficient approach utilizing abstract mean functions.

### 3.3 SCALABLE INFERENCE USING ABSTRACT MEAN FUNCTIONS

To create a model that is scalable in the number of modalities—a model that breaks the need for $2^M$ different networks—previous works define the joint posterior approximation $q_\phi(\boldsymbol{z}|\mathbb{X})$ as a mean function of the unimodal variational posteriors. The PoE and MoE can be subsumed under the concept of abstract means (Nielsen, 2019). Abstract means unify multiple mean functions $\mathcal{M}_f$ for a given function $f$ (Niculescu & Persson, 2005):

$$\mathcal{M}_f(\boldsymbol{p}) = f^{-1}\left(\frac{1}{P}\sum_{k=1}^{P} f(\boldsymbol{p}_k)\right)$$

where $P$ is the number of elements and the function $f$ needs to be injective in order for $f^{-1}$ to exist. $f(\boldsymbol{p}) = a\boldsymbol{p} + b$ results in the arithmetic mean, $f(\boldsymbol{p}) = \log \boldsymbol{p}$ in the geometric mean.

The choice of mean function directly influences the properties of the learned model as we will recapitulate with regard to multimodal VAEs in the following. The MVAE (Wu & Goodman, 2018) employs the PoE, which is a geometric mean of unimodal posteriors. Aggregation through the PoE results in a sharp posterior approximation (Hinton, 2002), but struggles in optimizing the individual experts as mentioned by the authors (Wu & Goodman, 2018, p. 3). In contrast, the MMVAE (Shi et al., 2019) uses the MoE, which is an arithmetic mean of unimodal posteriors. As such, the MMVAE optimizes individual experts well, but is not able to learn a distribution that is sharper than any of its experts. Thus, the choice of mean function directly influences the properties of the resulting model. The MoE is optimizing for conditional distributions based on the unimodal posterior approximations, while the PoE is optimizing for the approximation of the joint probability distribution.

For scalable, abstract-mean based models, the set of parameters $\phi_k$ for the posterior approximation of a subset $\tilde{q}_\phi(\boldsymbol{z}|\mathbb{X}_k)$ is determined by the unimodal posterior approximations $q_{\phi_j}(\boldsymbol{z}|\boldsymbol{x}_j)$ as $\phi_k = \{\phi_j \,\forall\, j \in \{1, \ldots, M\} : \boldsymbol{x}_j \in \mathbb{X}_k\}$.

### 3.4 GENERALIZED MULTIMODAL ELBO

In the following, we first introduce the new ELBO $\mathcal{L}_{\mathrm{MoPoE}}(\theta, \phi; \mathbb{X})$ and then prove that its objective minimizes the convex combination of KL-divergences in Equation (5).

---

[1]We omit the subscript $k$ for the parameterization of the posterior approximations when it is clear from context that only $\mathbb{X}_k$ is available, and write $\phi$ instead.

**Definition 1.**

1. *Let the posterior approximation of subset $\mathbb{X}_k$ be*

$$\tilde{q}_\phi(\boldsymbol{z}|\mathbb{X}_k) = PoE(\{q_{\phi_j}(\boldsymbol{z}|\boldsymbol{x}_j) \,\forall\, \boldsymbol{x}_j \in \mathbb{X}_k\}) \propto \prod\nolimits_{\boldsymbol{x}_j \in \mathbb{X}_k} q_{\phi_j}(\boldsymbol{z}|\boldsymbol{x}_j) \,.$$

2. *Let the joint posterior be $q_\phi(\boldsymbol{z}|\mathbb{X}) = \frac{1}{2^M} \sum_{\mathbb{X}_k \in \mathcal{P}(\mathbb{X})} \tilde{q}_\phi(\boldsymbol{z}|\mathbb{X}_k) \,.$*

*The objective $\mathcal{L}_{MoPoE}(\theta, \phi; \mathbb{X})$ for learning a joint distribution of multiple data types $\mathbb{X}$ is defined as*

$$\mathcal{L}_{MoPoE}(\theta, \phi; \mathbb{X}) := \mathbb{E}_{q_\phi(\boldsymbol{z}|\mathbb{X})}[\log(p_\theta(\mathbb{X}|\boldsymbol{z})] - D_{\mathrm{KL}}\Big(\frac{1}{2^M} \sum_{\mathbb{X}_k \in \mathcal{P}(\mathbb{X})} \tilde{q}_\phi(\boldsymbol{z}|\mathbb{X}_k)||p_\theta(\boldsymbol{z})\Big). \qquad (6)$$

From Definition 1, Lemma 2 directly follows.

**Lemma 2.** *$\mathcal{L}_{MoPoE}(\theta, \phi; \mathbb{X})$ is a multimodal ELBO, that is $\log p_\theta(\mathbb{X}) \geq \mathcal{L}_{MoPoE}(\theta, \phi; \mathbb{X})$.*

Since $q_\phi(\boldsymbol{z}|\mathbb{X})$ is defined as a mixture distribution (i.e., a probability distribution), it directly follows that $\mathcal{L}_{\mathrm{MoPoE}}(\theta, \phi; \mathbb{X})$ is a valid ELBO on $\log p_\theta(\mathbb{X})$, because a variational distribution can be chosen arbitrarily as long as it is a valid probability distribution. For a proof of Lemma 2, see Appendix A.2.

**Lemma 3.** *Maximizing $\mathcal{L}_{MoPoE}(\theta, \phi; \mathbb{X})$ minimizes the convex combination of KL-divergences of the powerset $\mathcal{P}(\mathbb{X})$ given in Equation (5).*

For a proof of Lemma 3, see Appendix A.3. Definition 1 does not put any restrictions on the choice of posterior approximations $\tilde{q}_\phi(\boldsymbol{z}|\mathbb{X}_k)$. As we are interested in scalable, multimodal models, we focus on methods which apply to this restriction and choose the PoE for the posterior approximations of the subsets $\mathbb{X}_k \in \mathcal{P}(\mathbb{X})$. Other, non-scalable posterior fusion methods are possible using this framework.

## 3.5 THE GENERAL FRAMEWORK

Definition 1 can be interpreted as a hierarchical distribution: first the unimodal posterior approximations of a subset $q_{\phi_j}(\boldsymbol{z}|\boldsymbol{x}_j) \,\forall\, \boldsymbol{x}_j \in \mathbb{X}_k$ are combined using a PoE, second the subset approximations $\tilde{q}_\phi(\boldsymbol{z}|\mathbb{X}_k) \,\forall\, \mathbb{X}_k \in \mathcal{P}(\mathbb{X})$ are combined using a MoE. This allows us to combine the strengths of both MoE as well as PoE while circumventing their weaknesses (see Section 2). For Gaussian posterior approximations, as is common in VAEs, the PoE can be calculated in closed form, which makes it a computationally efficient solution.

In the following, we derive the objectives optimized by the MVAE and MMVAE as special cases of $\mathcal{L}_{\mathrm{MoPoE}}(\theta, \phi; \mathbb{X})$. The MVAE only takes into account the full subset, i.e., the PoE of all data types. Trivially, this is a MoE with only a single component:

$$\mathcal{L}_{\mathrm{PoE}}(\theta, \phi; \mathbb{X}) = E_{q_\phi(\boldsymbol{z}|\mathbb{X})}[\log(p_\theta(\mathbb{X}|\boldsymbol{z})] - D_{\mathrm{KL}}(q_\phi(\boldsymbol{z}|\mathbb{X})||p_\theta(\boldsymbol{z})) \qquad (7)$$

$$\text{with } q_\phi(\boldsymbol{z}|\mathbb{X}) \propto \prod_{j=1}^{M} q_{\phi_j}(\boldsymbol{z}|\boldsymbol{x}_j) = PoE(\{q_{\phi_j}(\boldsymbol{z}|\boldsymbol{x}_j)\}_{j=1}^{M}) = \sum_{k=1}^{1} PoE(\{q_{\phi_j}(\boldsymbol{z}|\boldsymbol{x}_j)\}_{j=1}^{M}) \qquad (8)$$

This is equivalent to the MoPoE-VAE of a single subset $\mathbb{X}_K$, which is the full set $\mathbb{X}$.

As the PoE of a single expert is just the expert itself, the MMVAE model (Shi et al., 2019) is the special case of $\mathcal{L}_{\mathrm{MoPoE}}(\theta, \phi; \mathbb{X})$ which takes only into account the $M$ unimodal subsets:

$$\mathcal{L}_{\mathrm{MoE}}(\theta, \phi; \mathbb{X}) = E_{q_\phi(\boldsymbol{z}|\mathbb{X})}[\log(p_\theta(\mathbb{X}|\boldsymbol{z})] - D_{\mathrm{KL}}\left(\frac{1}{M} \sum_{j=1}^{M} q_{\phi_j}(\boldsymbol{z}|\boldsymbol{x}_j)||p_\theta(\boldsymbol{z})\right) \qquad (9)$$

$$\text{with } q_\phi(\boldsymbol{z}|\mathbb{X}) = \frac{1}{M} \sum_{j=1}^{M} q_{\phi_j}(\boldsymbol{z}|\boldsymbol{x}_j) = \frac{1}{M} \sum_{j=1}^{M} PoE(q_{\phi_j}(\boldsymbol{z}|\boldsymbol{x}_j)) \qquad (10)$$

$\mathcal{L}_{MoE}(\theta, \phi; \mathbb{X})$ is equivalent to a MoPoE-VAE of the $M$ unimodal posterior approximations $q_{\phi_j}(\boldsymbol{z}|\boldsymbol{x}_j)$ for $j = 1, \ldots, M$.

Therefore, the proposed MoPoE-VAE is a generalized formulation of the MVAE and MMVAE, which accounts for all subsets of modalities. The identified special cases offer a new perspective on the strengths and weaknesses of prior work: previous models focus on a specific subset of posteriors, which might lead to a decreased performance on the remaining subsets.

In particular, the MVAE should perform best when all modalities are present, whereas the MMVAE should be most suitable when only a single modality is observed. We validate this observation empirically in Section 4.

## 4 EXPERIMENTS & RESULTS

We evaluate the proposed method on three different datasets and compare it to state-of-the-art methods. We introduce a new dataset called PolyMNIST with 5 simplified modalities. Additionally, we evaluate all models on the trimodal matching digits dataset MNIST-SVHN-Text and the challenging bimodal Celeba dataset with images and text. The latter two were introduced in Sutter et al. (2020).

We evaluate the models according to three different metrics. We assess the quality of the learned latent representation using a linear classifier. The coherence of generated samples is evaluated using pre-trained classifiers. The approximation of the joint data distribution is measured using test set log-likelihoods.

The datasets and the evaluation of experiments are described in detail in Appendix B.

### 4.1 MNIST-SVHN-TEXT

Based on the MNIST-SVHN dataset (Shi et al., 2019), this trimodal dataset with an additional text modality forces a model to adapt to multiple data types. It involves data types of various difficulties. Whereas MNIST (LeCun & Cortes, 2010) and text are clean modalities, SVHN (Netzer et al., 2011) is comprised of noisy images.

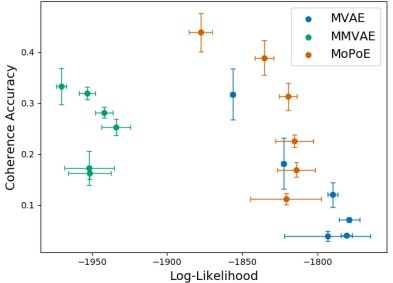

Figure 1: Joint Coherence vs. Log-Likelihoods for MNIST-SVHN-Text.

Tables 2 and 3 show the superior performance of the proposed method compared to state-of-the-art methods regarding the ability to learn meaningful latent representations and generate coherent samples. MVAE reaches superior performance for the generation of the SVHN modality, while MoPoE-VAE overall achieves best coherence results. Table 4 shows the results for the test log-likelihoods.

The proposed MoPoE-VAE is the only method that is able to reach state-of-the-art coherence, latent classification accuracies, as well as test log-likelihoods for all combination of inputs. This can be seen in Figure 1, illustrating the trade-off between test log-likelihoods and joint coherence for every model. Every point encodes the joint coherence and joint log-likelihood for a different $\beta$-value.[2] The goal is to have high coherence and log-likelihoods (i.e., the top right corner). Note that lower beta values typically correspond to models with a higher log-likelihood but lower coherence. Overall, the MoPoE-VAE achieves a superior trade-off compared to the baselines. As expected by our theoretical analysis (Section 3.5), the MVAE achieves good joint log-likelihoods, whereas MMVAE reaches high joint coherence.

---

[2] The $\beta$-hyperparameter controls the weight of the KL-divergence in Equation (6). We evaluate the models using $\beta \in \{0.5, 1.0, 2.5, 5.0, 10.0, 20.0\}$.

Table 2: Linear classification accuracy of latent representations for MNIST-SVHN-Text. We evaluate all subsets of modalities $\mathbb{X}_k$ where the abbreviations of subsets are as follows: M: MNIST; S: SVHN; T: Text; M,S: MNIST and SVHN; M,T: MNIST and Text; S,T: SVHN and Text; M,S,T: all. We report the means and standard deviations over 5 runs.

| MODEL | M | S | T | M,S | M,T | S,T | M,S,T |
|---|---|---|---|---|---|---|---|
| MVAE | $0.90_{\pm0.01}$ | $0.44_{\pm0.01}$ | $0.85_{\pm0.10}$ | $0.89_{\pm0.01}$ | $0.97_{\pm0.02}$ | $0.81_{\pm0.09}$ | $0.96_{\pm0.02}$ |
| MMVAE | $\mathbf{0.95}_{\pm0.01}$ | $0.79_{\pm0.05}$ | $\mathbf{0.99}_{\pm0.01}$ | $0.87_{\pm0.03}$ | $0.93_{\pm0.03}$ | $0.84_{\pm0.04}$ | $0.86_{\pm0.03}$ |
| MoPoE | $\mathbf{0.95}_{\pm0.01}$ | $\mathbf{0.80}_{\pm0.03}$ | $\mathbf{0.99}_{\pm0.01}$ | $\mathbf{0.97}_{\pm0.01}$ | $\mathbf{0.98}_{\pm0.01}$ | $\mathbf{0.99}_{\pm0.01}$ | $\mathbf{0.98}_{\pm0.01}$ |

Table 3: Generation coherence for MNIST-SVHN-Text. For conditional generation, the letter above the horizontal line indicates the modality which is generated based on the subsets $\mathbb{X}_k$ below. We report the mean values over 5 runs. Standard deviations are included in Appendix C.3.

| MODEL | JOINT | M | | | S | | | T | | |
|---|---|---|---|---|---|---|---|---|---|---|
| | | S | T | S,T | M | T | M,T | M | S | M,S |
| MVAE | 0.12 | 0.24 | 0.20 | 0.32 | **0.43** | 0.30 | **0.75** | 0.28 | 0.17 | 0.29 |
| MMVAE | 0.28 | **0.75** | **0.99** | 0.87 | 0.31 | 0.30 | 0.30 | **0.96** | 0.76 | 0.84 |
| MoPoE | **0.31** | 0.74 | **0.99** | **0.94** | 0.36 | **0.34** | 0.37 | **0.96** | 0.76 | **0.93** |

## 4.2 POLYMNIST

The PolyMNIST dataset consists of sets of MNIST digits where each set $\{x_j\}_{j=1}^{M}$ consists of 5 images with the same digit label but different backgrounds and different styles of hand writing. An example of one such tuple is shown in Figure 2. Thus, each "modality" represents a shuffled set of MNIST digits overlaid on top of (random crops from) 5 different background images, which are modality-specific. In total there are $60,000$ tuples of training examples and $10,000$ tuples of test examples and we make sure that no two MNIST digits were used in both the training and test set.[3]

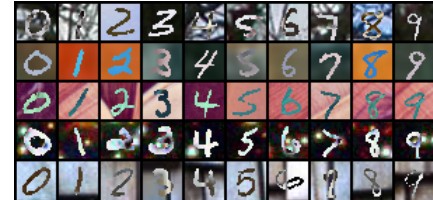

Figure 2: Ten samples from the PolyMNIST dataset. Each column depicts one tuple that consists of five different "modalities".

The PolyMNIST dataset allows to investigate how well different methods perform given more than two modalities. Since individual images can be difficult to classify correctly (even for a human observer) one would expect multimodal models to aggregate information across multiple modalities. Further, this dataset facilitates the comparison of different models, because it removes the need for modality-specific architectures and hyperparameters. As such, for a fair comparison, we use the same architectures and hyperparameter values across all methods. We expect to see that both the MMVAE and our proposed method are able to aggregate the redundant digit information across different modalities, whereas the MVAE should not be able to benefit from an increasing number of modalities, because it does not aggregate unimodal posteriors. Further, we hypothesize that the MVAE will achieve the best generative performance when all modalities are present, but that it will struggle with an increasing number of *missing* modalities. The proposed MoPoE-VAE should perform well given any subset of modalities.

**PolyMNIST results** Figure 3 compares the results across different methods. The performance in terms of three different metrics is shown as a function of the number of input modalities; for instance, the log-likelihood of all generated modalities given one input modality (averaged over all possible single input modalities). As expected, both the MVAE and MoPoE-VAE benefit from more input modalities, whereas the performance of the MVAE stays flat across all metrics. In the limit of all 5 input modalities, the log-likelihood of MoPoE-VAE is on par with MVAE, but the proposed method is clearly superior in terms of both latent classification as well as conditional coherence

---

[3]Details on how the dataset was generated are included in Appendix D.

Table 4: Test set log-likelihoods on MNIST-SVHN-Text. We report the test set log-likelihoods of the joint generative model conditioned on the variational posterior of subsets of modalities $\tilde{q}_\phi(\boldsymbol{z}|\mathbb{X}_k)$. ($\boldsymbol{x}_M$: MNIST; $\boldsymbol{x}_S$: SVHN; $\boldsymbol{x}_T$: Text; $\mathbb{X} = (\boldsymbol{x}_M, \boldsymbol{x}_S, \boldsymbol{x}_T)$).

| MODEL | $\mathbb{X}$ | $\mathbb{X}|\boldsymbol{x}_M$ | $\mathbb{X}|\boldsymbol{x}_S$ | $\mathbb{X}|\boldsymbol{x}_T$ | $\mathbb{X}|\boldsymbol{x}_M,\boldsymbol{x}_S$ | $\mathbb{X}|\boldsymbol{x}_M,\boldsymbol{x}_T$ | $\mathbb{X}|\boldsymbol{x}_S,\boldsymbol{x}_T$ |
|---|---|---|---|---|---|---|---|
| MVAE | **-1790**$_{\pm 3.3}$ | -2090$_{\pm 3.8}$ | -1895$_{\pm 0.2}$ | -2133$_{\pm 6.9}$ | -1825$_{\pm 2.6}$ | -2050$_{\pm 2.6}$ | **-1855**$_{\pm 0.3}$ |
| MMVAE | -1941$_{\pm 5.7}$ | **-1987**$_{\pm 1.5}$ | **-1857**$_{\pm 12}$ | **-2018**$_{\pm 1.6}$ | -1912$_{\pm 7.3}$ | -2002$_{\pm 1.2}$ | -1925$_{\pm 7.7}$ |
| MoPoE | -1819$_{\pm 5.7}$ | -1991$_{\pm 2.9}$ | **-1858**$_{\pm 6.2}$ | -2024$_{\pm 2.6}$ | **-1822**$_{\pm 5.0}$ | **-1987**$_{\pm 3.1}$ | **-1850**$_{\pm 5.8}$ |

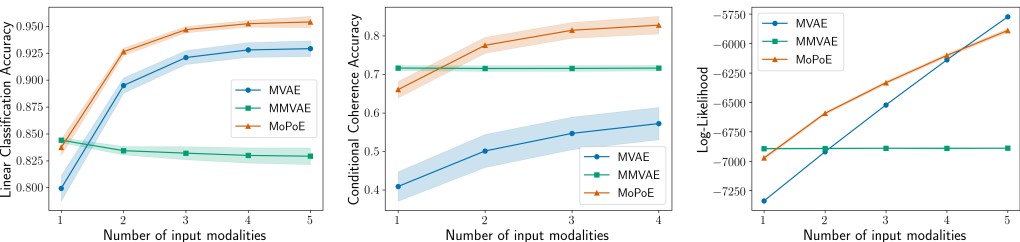

Figure 3: Performance on PolyMNIST as a function of the number of input modalities, averaged over all subsets of the respective size. Performance is measured in terms of three different metrics (larger is better) and markers denote the means (error bands denote standard deviations) over five runs. *Left:* Linear classification accuracy of digits given the latent representation computed from the respective subset. *Center:* Coherence of conditionally generated samples (excluding the input modality). *Right:* Log-likelihood of all generated modalities. *Not shown:* The joint coherence is $3.6\,(\pm 1.5)$, $20.0\,(\pm 1.9)$, and $12.1\,(\pm 1.6)$ percent for MVAE, MMVAE, and MoPoE respectively.

across any subset of modalities. Analogously, in the limit of a single input modality, MoPoE-VAE matches the performance of MMVAE. Only in terms of the joint coherence (see figure legend) the MMVAE performs better, suggesting that a more flexible prior might be needed for the MoPoE-VAE. Therefore, the PolyMNIST experiment illustrates that the proposed method does not only theoretically encompass the other two methods, but that it is superior for most subsets of modalities and even matches the performance in special cases that favor previous methods.

## 4.3 BIMODAL CELEBA

In this dataset, the images displaying faces (Liu et al., 2015) are equipped with additional text describing the faces using the labeled attributes. Any negatively labeled attribute is completely missing in the string which makes the text modality more challenging. Compared to previous experiments, we additionally use modality-specific latent spaces, which were found to improve the generative quality of a model (Hsu & Glass, 2018; Sutter et al., 2020; Daunhawer et al., 2020).[4] Figure 4 displays qualitative results of images which are generated given text. Table 5 shows the classification results for the coherence of generated samples as well as the classification of latent representations. We see that the proposed model is able to match the baselines on this challenging dataset, which favors the baselines, because it consists of two modalities. Figure 4 shows that attributes like "gender" or "smiling" are learned well, as they manifest in generated samples and can be identified from the latent representation. Subtle and rare attributes are more difficult to generate consistently; evaluations specific to the different labels are provided in Appendix E.3.

## 5 CONCLUSION

In this work, we propose a new multimodal ELBO formulation. Our contribution is threefold: First, the proposed MoPoE-VAE generalizes prior works (MVAE, MMVAE) and combines their benefits. Second, we analyze the strengths and weaknesses of previous works and relate them directly to their

---

[4]For more details, see Appendix E.

Table 5: Classification and coherence results on the bimodal CelebA experiment. For latent representations and conditionally generated samples, we report the mean average precision over all attributes (I: Image; T: Text; Joint: I and T).

| | LATENT REPRESENTATION | | | GENERATION | |
|---|---|---|---|---|---|
| MODEL | I | T | JOINT | I → T | T → I |
| MVAE | 0.30 | 0.31 | 0.32 | **0.26** | 0.33 |
| MMVAE | 0.35 | 0.38 | 0.35 | 0.14 | 0.41 |
| MoPoE | **0.40** | **0.39** | **0.39** | 0.15 | **0.43** |

Figure 4: Qualitative results for bimodal CelebA. The images are conditionally generated by MoPoE-VAE using the text on top of each column.

objective and choice of posterior approximation function. Finally, in extensive experiments we empirically show the advantages compared to state-of-the-art models and even match their performance on tasks that favor previous work. In future work, we would like to evaluate previous extensions to multimodal VAEs. Addtionally, we will explore different types and combinations of abstract mean functions and investigate their effects on the model and its performance as well as their theoretical properties (e.g., tightness) compared to existing methods.

ACKNOWLEDGMENTS

We would like to thank Ričards Marcinkevičs for helpful discussions and proposing the name "PolyMNIST". ID is supported by the SNSF grant #200021_188466.

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

# A  PROOFS

## A.1  PROOF OF LEMMA 1

$\sum_{\mathbb{X}_k \in \mathcal{P}(\mathbb{X})} D_{\mathrm{KL}}(\tilde{q}_\phi(\boldsymbol{z}|\mathbb{X}_k)||p_\theta(\boldsymbol{z}|\mathbb{X}))$, *i.e., the sum of KL-divergences in Equation* (5) *can be used for describing the joint probability* $\log p_\theta(\mathbb{X})$.

*Proof.* We show that the convex combination of KL-divergences can be directly related to the joint probability $\log p_\theta(\mathbb{X})$:

$$\sum_{\mathbb{X}_k \in \mathcal{P}(\mathbb{X})} D_{\mathrm{KL}}(\tilde{q}_\phi(\boldsymbol{z}|\mathbb{X}_k)||p_\theta(\boldsymbol{z}|\mathbb{X})) = \sum_{\mathbb{X}_k \in \mathcal{P}(\mathbb{X})} \mathbb{E}_{\tilde{q}_\phi(\boldsymbol{z}|\mathbb{X}_k)} \left[ \log \frac{\tilde{q}_\phi(\boldsymbol{z}|\mathbb{X}_k)}{p_\theta(\boldsymbol{z}|\mathbb{X})} \right]$$

$$= \sum_{\mathbb{X}_k \in \mathcal{P}(\mathbb{X})} \left( \mathbb{E}_{\tilde{q}_\phi(\boldsymbol{z}|\mathbb{X}_k)} \left[ \log \frac{\tilde{q}_\phi(\boldsymbol{z}|\mathbb{X}_k)}{p_\theta(\boldsymbol{z},\mathbb{X})} \right] + \log p_\theta(\mathbb{X}) \right) \quad (11)$$

which can be reformulated as an expression of the joint probability $\log p_\theta(\mathbb{X})$:

$$\log p_\theta(\mathbb{X}) = \frac{1}{2^M} \underbrace{\sum_{\mathbb{X}_k \in \mathcal{P}(\mathbb{X})} D_{\mathrm{KL}}(\tilde{q}_\phi(\boldsymbol{z}|\mathbb{X}_k)||p_\theta(\boldsymbol{z}|\mathbb{X}))}_{Equation~(5)} + \frac{1}{2^M} \sum_{\mathbb{X}_k \in \mathcal{P}(\mathbb{X})} \mathbb{E}_{\tilde{q}_\phi(\boldsymbol{z}|\mathbb{X}_k)} \left[ \log \frac{p_\theta(\mathbb{X}|\boldsymbol{z})p_\theta(\boldsymbol{z})}{\tilde{q}_\phi(\boldsymbol{z}|\mathbb{X}_k)} \right]$$

From the non-negativity of the KL-divergence, we derive the lower bound to the joint probability $\log p(\mathbb{X})$:

$$\log p_\theta(\mathbb{X}) \geq \frac{1}{2^M} \sum_{\mathbb{X}_k \in \mathcal{P}(\mathbb{X})} \mathbb{E}_{\tilde{q}_\phi(\boldsymbol{z}|\mathbb{X}_k)} \left[ \log \frac{p_\theta(\mathbb{X}|\boldsymbol{z})p_\theta(\boldsymbol{z})}{\tilde{q}_\phi(\boldsymbol{z}|\mathbb{X}_k)} \right] \quad (12)$$

$\square$

## A.2  PROOF OF LEMMA 2

$\mathcal{L}_{MoPoE}(\theta, \phi; \mathbb{X})$ *is a multimodal ELBO, that is* $\log p_\theta(\mathbb{X}) \geq \mathcal{L}_{MoPoE}(\theta, \phi; \mathbb{X})$.

*Proof.* The sums using index $k$ also sum over all $2^M$ subsets in the power set $\mathcal{P}(\mathbb{X})$. We use $k$ only for better readability.

$$\log p(\mathbb{X}) = D_{\mathrm{KL}}(q_\phi(\boldsymbol{z}|\mathbb{X})||p(\boldsymbol{z}|\mathbb{X})) + E_{q_\phi(\boldsymbol{z}|\mathbb{X})}[\log \frac{p(\boldsymbol{z},\mathbb{X})}{q_\phi(\boldsymbol{z}|\mathbb{X})}] \quad (13)$$

$$= D_{\mathrm{KL}} \left( \frac{1}{2^M} \sum_k \tilde{q}(\boldsymbol{z}|\mathbb{X}_k)||p(\boldsymbol{z}|\mathbb{X}) \right) + E_{\frac{1}{2^M} \sum_k \tilde{q}(\boldsymbol{z}|\mathbb{X}_k)} \left[ \log \frac{p(\boldsymbol{z},\mathbb{X})}{\frac{1}{2^M} \sum_k \tilde{q}(\boldsymbol{z}|\mathbb{X}_k)} \right] \quad (14)$$

$$\geq E_{\frac{1}{2^M} \sum_k \tilde{q}(\boldsymbol{z}|\mathbb{X}_k)} \left[ \log \frac{p(\boldsymbol{z},\mathbb{X})}{\frac{1}{2^M} \sum_k \tilde{q}(\boldsymbol{z}|\mathbb{X}_k)} \right] \quad (15)$$

$$= E_{\tilde{q}_\phi(\boldsymbol{z}|\mathbb{X})}[\log(p_\theta(\mathbb{X}|\boldsymbol{z}) - D_{\mathrm{KL}}(\frac{1}{2^M} \sum_{\mathbb{X}_k \in \mathcal{P}(\mathbb{X})} \tilde{q}_{\phi_k}(\boldsymbol{z}|\mathbb{X}_k)||p_\theta(\boldsymbol{z})) \quad (16)$$

$$= \mathcal{L}(\theta, \phi; \mathbb{X}) \quad (17)$$

$\square$

## A.3  PROOF OF LEMMA 3

Maximizing $\mathcal{L}_{MoPoE}(\theta, \phi; \mathbb{X})$ *minimizes the convex combination of KL-divergences of the powerset* $\mathcal{P}(\mathbb{X})$ *given in Equation* (5).

*Proof.* Lemma 1 shows that the sum of KL-divergences in Equation (5) is able to describe the joint probability and can be used to form a valid ELBO. Equation (12) is the convex combination of ELBOs given a subset's posterior approximation $\tilde{q}_\phi(\boldsymbol{z}|\mathbb{X}_k)$. Utilizing Jensen's inequality, it follows:

$$\frac{1}{2^M} \sum_{\mathbb{X}_k \in \mathcal{P}(\mathbb{X})} \mathbb{E}_{\tilde{q}_\phi(\boldsymbol{z}|\mathbb{X}_k)} \left[ \log \frac{p_\theta(\mathbb{X}|\boldsymbol{z})p_\theta(\boldsymbol{z})}{\tilde{q}_\phi(\boldsymbol{z}|\mathbb{X}_k)} \right] \leq \mathbb{E}_{\frac{1}{2^M}\sum_k \tilde{q}_\phi(\boldsymbol{z}|\mathbb{X}_k)} \left[ \log \frac{p_\theta(\mathbb{X}|\boldsymbol{z})p_\theta(\boldsymbol{z})}{\frac{1}{2^M}\sum_k \tilde{q}_\phi(\boldsymbol{z}|\mathbb{X}_k)} \right] \quad (18)$$

where the sums on the right hand also iterate over all $\mathbb{X}_k \in \mathcal{P}(\mathbb{X})$. From Equation (18), we see that the proposed $\mathcal{L}_{\text{MoPoE}}(\theta, \phi; \mathbb{X})$ is not only a valid lower bound to the joint log-probability $\log p_\theta(\mathbb{X})$, but also a tighter one than the convex combination of ELBOs:

$$\log p_\theta(\mathbb{X}) \geq \mathbb{E}_{\frac{1}{2^M}\sum_k \tilde{q}_\phi(\boldsymbol{z}|\mathbb{X}_k)} \left[ \log \frac{p_\theta(\mathbb{X}|\boldsymbol{z})p_\theta(\boldsymbol{z})}{\frac{1}{2^M}\sum_k \tilde{q}_\phi(\boldsymbol{z}|\mathbb{X}_k)} \right] \quad (19)$$

$$= \mathcal{L}_{\text{MoPoE}}(\theta, \phi; \mathbb{X}) \quad (20)$$

$$\geq \frac{1}{2^M} \sum_{\mathbb{X}_k \in \mathcal{P}(\mathbb{X})} \mathbb{E}_{\tilde{q}_\phi(\boldsymbol{z}|\mathbb{X}_k)} \left[ \log \frac{p_\theta(\mathbb{X}|\boldsymbol{z})p_\theta(\boldsymbol{z})}{\tilde{q}_\phi(\boldsymbol{z}|\mathbb{X}_k)} \right] \quad (21)$$

As Equation (19) can be directly derived from Equation (5), maximizing the proposed objective results in minimizing the convex combination of KL-divergences.

In the following, we derive the inequality in Equation (18) in more detail:

$$\frac{1}{2^M} \sum_{\mathbb{X}_k \in \mathcal{P}(\mathbb{X})} \mathbb{E}_{\tilde{q}_\phi(\boldsymbol{z}|\mathbb{X}_k)} \left[ \log \frac{p_\theta(\mathbb{X}|\boldsymbol{z})p_\theta(\boldsymbol{z})}{\tilde{q}_\phi(\boldsymbol{z}|\mathbb{X}_k)} \right] \quad (22)$$

$$= \frac{1}{2^M} \sum_{\mathbb{X}_k \in \mathcal{P}(\mathbb{X})} \mathbb{E}_{\tilde{q}_\phi(\boldsymbol{z}|\mathbb{X}_k)} \left[ \log p_\theta(\mathbb{X}|\boldsymbol{z})p_\theta(\boldsymbol{z}) - \log \tilde{q}_\phi(\boldsymbol{z}|\mathbb{X}_k) \right] \quad (23)$$

$$= \underbrace{\frac{1}{2^M} \sum_{\mathbb{X}_k \in \mathcal{P}(\mathbb{X})} \mathbb{E}_{\tilde{q}_\phi(\boldsymbol{z}|\mathbb{X}_k)} \left[ \log p_\theta(\mathbb{X}|\boldsymbol{z})p_\theta(\boldsymbol{z}) \right]}_{=\mathbb{E}_{\frac{1}{2^M}\sum_k \tilde{q}_\phi(\boldsymbol{z}|\mathbb{X}_k)}[\log p_\theta(\mathbb{X}|\boldsymbol{z})p_\theta(\boldsymbol{z})]} - \underbrace{\frac{1}{2^M} \sum_{\mathbb{X}_k \in \mathcal{P}(\mathbb{X})} \mathbb{E}_{\tilde{q}_\phi(\boldsymbol{z}|\mathbb{X}_k)} \left[ \log \tilde{q}_\phi(\boldsymbol{z}|\mathbb{X}_k) \right]}_{\geq \mathbb{E}_{\frac{1}{2^M}\sum_k \tilde{q}_\phi(\boldsymbol{z}|\mathbb{X}_k)}\left[\log\left(\frac{1}{2^M}\sum_k \tilde{q}_\phi(\boldsymbol{z}|\mathbb{X}_k)\right)\right]} \quad (24)$$

$$\leq \mathbb{E}_{\frac{1}{2^M}\sum_k \tilde{q}_\phi(\boldsymbol{z}|\mathbb{X}_k)} \left[ \log p_\theta(\mathbb{X}|\boldsymbol{z})p_\theta(\boldsymbol{z}) \right] - \mathbb{E}_{\frac{1}{2^M}\sum_k \tilde{q}_\phi(\boldsymbol{z}|\mathbb{X}_k)} \left[ \log \left( \frac{1}{2^M} \sum_k \tilde{q}_\phi(\boldsymbol{z}|\mathbb{X}_k) \right) \right] \quad (25)$$

$$= \mathbb{E}_{\frac{1}{2^M}\sum_k \tilde{q}_\phi(\boldsymbol{z}|\mathbb{X}_k)} \left[ \log \left( p_\theta(\mathbb{X}|\boldsymbol{z})p_\theta(\boldsymbol{z}) \right) - \log \left( \frac{1}{2^M} \sum_k \tilde{q}_\phi(\boldsymbol{z}|\mathbb{X}_k) \right) \right] \quad (26)$$

$$= \mathbb{E}_{\frac{1}{2^M}\sum_k \tilde{q}_\phi(\boldsymbol{z}|\mathbb{X}_k)} \left[ \log \frac{p_\theta(\mathbb{X}|\boldsymbol{z})p_\theta(\boldsymbol{z})}{\frac{1}{2^M}\sum_k \tilde{q}_\phi(\boldsymbol{z}|\mathbb{X}_k)} \right] \quad (27)$$

$\square$

In the minuend of Equation (24), the ordering of expectation and sum can be exchanged due to the linearity of the expectation. In the subtrahend of Equation (24), the sum of expectation of the posterior approximations of subsets can be reformulated into the expectation of a mixture distribution using Jensen's inequality. Due to the convexity of the function $f(t) = t \log t$ (Cover & Thomas, 2006, p.29), the expectation of a mixture distribution is a lower bound to the sum over the expectation of posterior approximations as the mixture distribution can be seen as a convex combination of posterior approximations of subset of modalities $\tilde{q}_\phi(\boldsymbol{z}|\mathbb{X}_k)$. Hence, the inequality from Equation (24) to Equation (25) follows as we decrease the subtrahend in Equation (25).

## B  EVALUATION OF EXPERIMENTS

For the experiments, we evaluate all models regarding three different metrics: the classification accuracy (or average precision for CelebA) on the latent representation, the coherence of generated samples and the test set log-likelihoods.

The latent representations are evaluated using a logistic regression classifier from scikit-learn (Pedregosa et al., 2011). The classifier is trained using 500 samples from the training set which are encoded using the trained models. The evaluation is done on the full test set and the reported numbers are the average performances over all batches in the test set.

The generation coherence is evaluated using the same networks as the unimodal encoders which were trained beforehand. For every data type, we train a neural network classifier in a supervised way. The architecture of the classifier is identical to the encoder except from the last layer. For joint coherence, all generated samples are evaluated by the classifier and if all modalities are classified as having the same label, they are considered coherent. The coherence accuracy is the ratio of coherent samples divided by the number of generated samples. For conditional generation, the conditionally generated samples have to be coherent to the input samples.

The test set log-likelihoods are evaluated using 15 importance samples for all models and the reported numbers are the averages over all test set batches.

If not stated differently, the reported numbers in section 4 are the mean and standard deviations of 5 runs with different random seeds. All models evaluated use the same architectures and numbers of parameters. The likelihoods of the different modalities are weighted to each other according to the size of the modality for all experiments. The most dominant modality is set to $1.0$. The remaining ones are scaled up by the ratio of their data dimensions. For example in the MNIST-SVHN-Text experiment, SVHN is set to $1.0$ and MNIST to $3.92$ which is the ratio of their data dimensions.

For all unimodal posterior approximations, we assume Gaussian distributions $\mathcal{N}(\boldsymbol{z}; \boldsymbol{\mu}, \boldsymbol{\sigma}^2 \boldsymbol{I}_n)$ where $n$ is the number of latent space dimensions. In all experiments, the mixture components are equally weighted with $\frac{1}{\#components}$.

### B.1  COMPARISON TO PREVIOUS WORKS

Shi et al. (2019) in the end use a different ELBO objective including importance samples $\mathcal{L}_{\text{IWAE}}$ (Burda et al., 2016). We compare all models without the use of importance samples as these could be easily introduced to all objectives and are not directly related to the focus of this work which is choice of joint posterior approximation.

Sutter et al. (2020) utlize the Jensen-Shannon divergence as a regularizer instead of the KL-divergence. This results in the use of a dynamic prior and shows promising results. Besides the dynamic prior, they model the joint posterior approximation as well using a MoE. Again, we do not include models utilizing a dynamic prior as this could be introduced to all formulations and is not the focus of this work.

#### B.1.1  EQUIVALENCE TO ELBO FORMULATION IN SHI ET AL. (2019)

For clarity, we show here the equivalence of the formulation in Equation (10) to the formulation in (Shi et al., 2019, p.5).

$$\mathcal{L}_{\text{MoE}}(\theta, \phi; \mathbb{X}) = E_{q_\phi(\boldsymbol{z}|\mathbb{X})}[\log(p_\theta(\mathbb{X}|\boldsymbol{z})] - D_{\text{KL}}\left(\frac{1}{M}\sum_{j=1}^{M} q_{\phi_j}(\boldsymbol{z}|\boldsymbol{x}_j)\,||p_\theta(\boldsymbol{z})\right) \tag{28}$$

$$= E_{q_\phi(\boldsymbol{z}|\mathbb{X})}[\log(p_\theta(\mathbb{X}|\boldsymbol{z})] - E_{\frac{1}{M}\sum_{j=1}^{M} q_{\phi_j}(\boldsymbol{z}|\boldsymbol{x}_j)}\left[\log\frac{\frac{1}{M}\sum_{j=1}^{M} q_{\phi_j}(\boldsymbol{z}|\boldsymbol{x}_j)}{p_\theta(\boldsymbol{z})}\right] \tag{29}$$

$$= E_{q_\phi(\boldsymbol{z}|\mathbb{X})}[\log(p_\theta(\mathbb{X}|\boldsymbol{z})] - E_{q_\phi(\boldsymbol{z}|\mathbb{X})}\left[\log\frac{q_\phi(\boldsymbol{z}|\mathbb{X})}{p_\theta(\boldsymbol{z})}\right] \tag{30}$$

$$= E_{q_\phi(\boldsymbol{z}|\mathbb{X})}[\log p_\theta(\mathbb{X}|\boldsymbol{z})] + E_{q_\phi(\boldsymbol{z}|\mathbb{X})}\left[\log\frac{p_\theta(\boldsymbol{z})}{q_\phi(\boldsymbol{z}|\mathbb{X})}\right] \tag{31}$$

$$= E_{q_\phi(\boldsymbol{z}|\mathbb{X})}\left[\log\frac{p_\theta(\mathbb{X}|\boldsymbol{z})p_\theta(\boldsymbol{z})}{q_\phi(\boldsymbol{z}|\mathbb{X})}\right] \tag{32}$$

$$= \frac{1}{M}\sum_{j=1}^{M} E_{q_{\phi_j}(\boldsymbol{z}|\boldsymbol{x}_j)}\left[\log\frac{p_\theta(\mathbb{X}|\boldsymbol{z})p_\theta(\boldsymbol{z})}{q_\phi(\boldsymbol{z}|\mathbb{X})}\right] \tag{33}$$

where Equation (32) and Equation (33) are equivalent to the first equation on page 5 in Shi et al. The different formulation on the second line of the first equation on page 5 is coming from their use of importance samples.

Table 6: Generation Coherence for MNIST-SVHN-Text. For every subtable, the modality above the wide horizontal line is generated based on the subsets below the same line—except for joint coherence. The abbreviations of the different modalities are as follows: M:MNIST; S: SVHN; T: Text. Combinations thereof separated by commas result in the subsets consisting of the modalities. We report the mean value and standard deviation of 5 runs.

| | M | | |
|---|---|---|---|
| MODEL | S | T | S,T |
| MVAE | 0.24±0.01 | 0.20±0.05 | 0.32±0.03 |
| MMVAE | 0.75±0.06 | 0.99±0.01 | 0.87±0.03 |
| MoPoE | 0.74±0.04 | 0.99±0.01 | 0.94±0.01 |

| | S | | |
|---|---|---|---|
| MODEL | M | T | M,T |
| MVAE | 0.43±0.02 | 0.30±0.08 | 0.75±0.04 |
| MMVAE | 0.31±0.03 | 0.30±0.04 | 0.30±0.03 |
| MoPoE | 0.36±0.07 | 0.34±0.06 | 0.37±0.06 |

| | T | | |
|---|---|---|---|
| MODEL | M | S | M,S |
| MVAE | 0.28±0.06 | 0.17±0.02 | 0.29±0.06 |
| MMVAE | 0.96±0.01 | 0.76±0.04 | 0.84±0.02 |
| MoPoE | 0.96±0.01 | 0.76±0.03 | 0.93±0.01 |

| MODEL | JOINT COHERENCE |
|---|---|
| MVAE | 0.12±0.02 |
| MMVAE | 0.28±0.01 |
| MoPoE | 0.31±0.03 |

## C  MNIST-SVHN-Text

### C.1  Dataset

The dataset MNIST-SVHN-Text was introduced and described by Sutter et al. (2020). Equal to Shi et al. (2019) in their bimodal experiment, we create 20 triples per set resulting in a many-to-many mapping.

### C.2  Experimental Setup

The latent space dimension is set to 20 for all modalities, models and runs. The results in tables 2 to 4 are generated with $\beta = 5.0$. We train all models for 150 epochs. We use the same architectures as in Sutter et al. (2020). For MNIST encoder and decoder, we use fully-connected layers, for SVHN and text encoders and decoder feed-forward convolutional layers. For all layers, we use ReLU-activation functions (Nair & Hinton, 2010). The detailed architectures can also be looked up in the released code. We use an Adam optimizer (Kingma & Ba, 2014) with an initial learning rate 0.001.

### C.3  Additional Results

In table 6, we show the coherence results including the standard deviation of the 5 runs which were removed from the main part due to space restrictions.

Additionally, we perform the analysis of coherence in relation to log-likelihood for conditional generation as well, similar to the example using random generation in section 4.1. The combination of coherence and log-likelihoods shows the ability of a model to learn the data distribution as well as the generation of coherent samples. Every point refers to a different $\beta$ value. We evaluated the models for $\beta = [0.5, 1.0, 2.5, 5.0, 10.0, 20.0]$. The points in the figures are the mean values of 5 different runs with the lines being the standard deviations in both directions, coherence and log-likelihoods.

Figure 7 displays a qualitative comparison between the three models using 100 randomly generated samples. The generated samples correspond to the numbers in section 4.1. MVAE is able to best approximate the joint distribution in terms of sample quality for the price of a limited coherence,

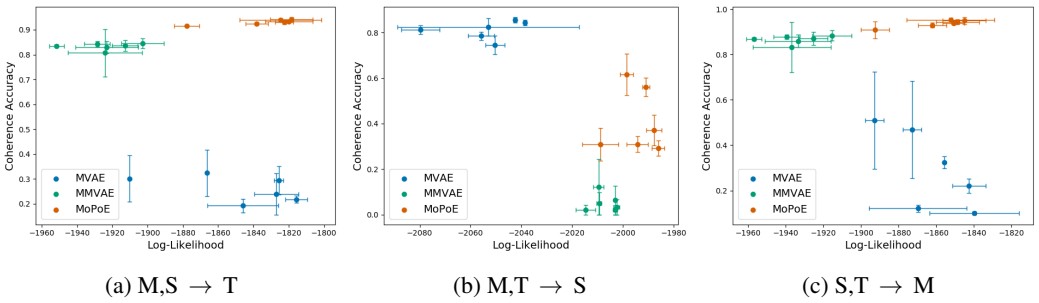

| (a) M,S → T | (b) M,T → S | (c) S,T → M |
| --- | --- | --- |

Figure 5: Coherence and Log-Likelihoods for MNIST-SVHN-Text. The three figures show the evaluation for the conditional generation of a single modality given the other two in relation to the joint log-likelihood given these two modalities, e.g. in the first row we generate SVHN samples conditioned on MNIST and Text. The points in the figures are the mean values of 5 different runs with the lines being the standard deviations in bopth directions, coherence and log-likelihoods.

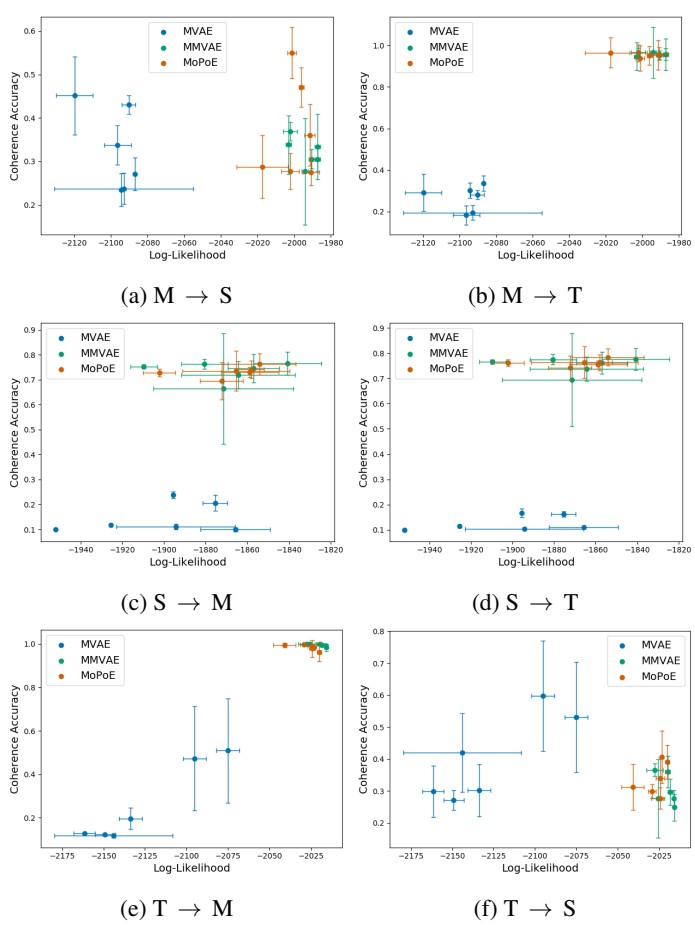

| (a) M → S | (b) M → T |
| --- | --- |
| (c) S → M | (d) S → T |
| (e) T → M | (f) T → S |

Figure 6: Coherence and Log-Likelihoods for MNIST-SVHN-Text. The three rows of figures show the evaluation for the conditional generation of two modalities given the remaining one in relation to the joint log-likelihood given this single modality, e.g. in the first row we generate SVHN and Text samples conditioned on MNIST. The points in the figures are the mean values of 5 different runs with the lines being the standard deviations in both directions, coherence and log-likelihoods.

while MMVAE shows higher coherence but limited sample quality. MoPoE approximate MVAE's sample quality with a start-of-the-art coherence.

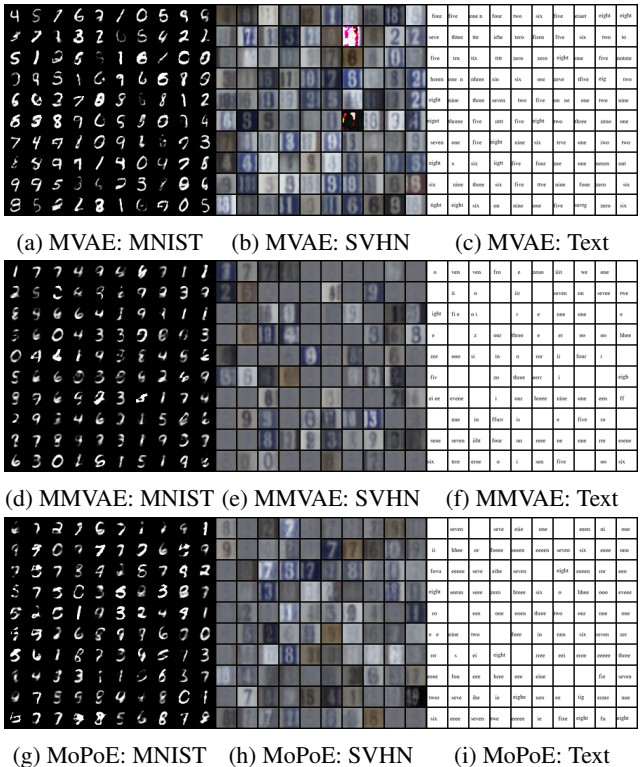

Figure 7: Qualitative comparison of randomly generate MNIST-SVHN-Text samples.

In addition to the theoretical proof of Lemma 3 that Definition 1 minimizes the convex combination of ELBOs, we compare the performance of a model trained using the objective in Equation (5) to the proposed method MoPoE-VAE. It can be seen that MoPoE-VAE achieves competitive results to the model which is optimizing Equation (5). This shows empirically that the proposed method is indeed minimizing the convex combination of ELBOs in Equation (5). Equation (5) is extensively minimizing the ELBO of every possible subset. Hence, Equation (5) is computationally much more expensive to optimize.

Table 7: Comparison of objectives: Equation (5) and Definition 1. We report the test set log-likelihoods of the joint generative model conditioned on the variational posterior of subsets of modalities $\tilde{q}_\phi(z|\mathbb{X}_k)$. ($\boldsymbol{x}_M$: MNIST; $\boldsymbol{x}_S$: SVHN; $\boldsymbol{x}_T$: Text; $\mathbb{X} = (\boldsymbol{x}_M, \boldsymbol{x}_S, \boldsymbol{x}_T)$). For both objectives we use $\beta = 2.5$

| MODEL | $\mathbb{X}$ | $\mathbb{X}|\boldsymbol{x}_M$ | $\mathbb{X}|\boldsymbol{x}_S$ | $\mathbb{X}|\boldsymbol{x}_T$ | $\mathbb{X}|\boldsymbol{x}_M, \boldsymbol{x}_S$ | $\mathbb{X}|\boldsymbol{x}_M, \boldsymbol{x}_T$ | $\mathbb{X}|\boldsymbol{x}_S, \boldsymbol{x}_T$ |
|---|---|---|---|---|---|---|---|
| EQ. (5) | -1810 | -1993 | -1831 | -2039 | -1811 | -2000 | -1839 |
| MOPOE | $-1815_{\pm 12.4}$ | $-1990_{\pm 4.4}$ | $-1858_{\pm 13.2}$ | $-2024_{\pm 1.2}$ | $-1819_{\pm 13.4}$ | $-1986_{\pm 2.5}$ | $-1848_{\pm 11.5}$ |

# D  POLYMNIST

## D.1  DATASET

For the creation of the PolyMNIST dataset, we fuse each MNIST image with a random crop of size 28x28 from the background image of the respective modality. In particular, we binarize the MNIST image and invert the colors of the random crop at those locations where the binarized MNIST digit is visible. We use the following background images:

1. John Burkardt.  Licensed under GNU LGPL. `https://people.sc.fsu.edu/~jburkardt/data/jpg/fractal_tree.jpg` [Online; retrieved 27.09.2020]

2. Edvard Munch. The Scream. Public domain. `https://upload.wikimedia.org/wikipedia/commons/f/f4/The_Scream.jpg` [Online; retrieved 27.09.2020]

3. The Waterloo Image Repository.  Lena.  Copyright belongs to the author. `http://links.uwaterloo.ca/Repository/TIF/lena3.tif` [Online; retrieved 27.09.2020]

4. John Burkardt.  Licensed under GNU LGPL. `https://people.sc.fsu.edu/~jburkardt/data/jpg/star_field.jpg` [Online; retrieved 27.09.2020]

5. John Burkardt.  Licensed under GNU LGPL. `https://people.sc.fsu.edu/~jburkardt/data/jpg/shingles.jpg` [Online; retrieved 27.09.2020]

## D.2  EXPERIMENTAL SETUP

The latent space dimension is set to 512 for all modalities, models and runs. All results in are based on $\beta = 2.5$, which was found to be a reasonable setting for all models. We use the same architectures for all methods and train all models for 300 epochs. We use an Adam optimizer (Kingma & Ba, 2014) with an initial learning rate 0.001. The architecture is based on straightforward convolutional neural networks (without bells and whistles); for details, we refer to the released code.

## D.3  QUALITATIVE RESULTS

In Figures 8 to 11, we show qualitative results comparing the different methods.

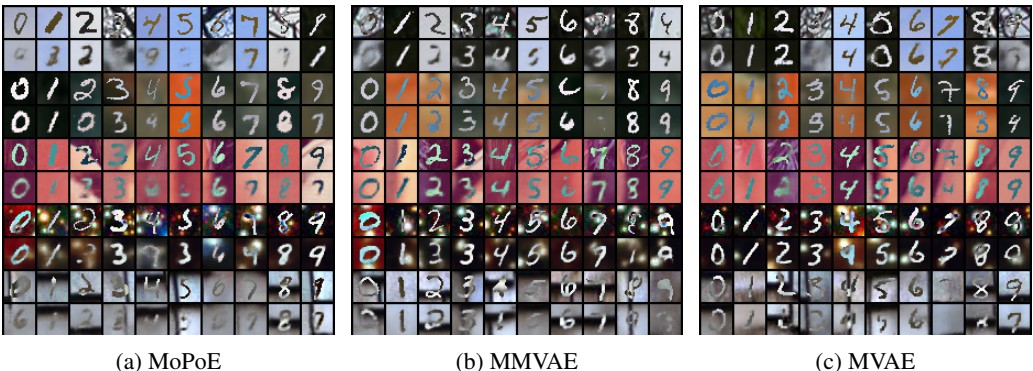

     (a) MoPoE            (b) MMVAE            (c) MVAE

Figure 8: Reconstructions across all modalities for all models. In every pair of rows, we show one row of test images followed by one row of respective reconstructions.

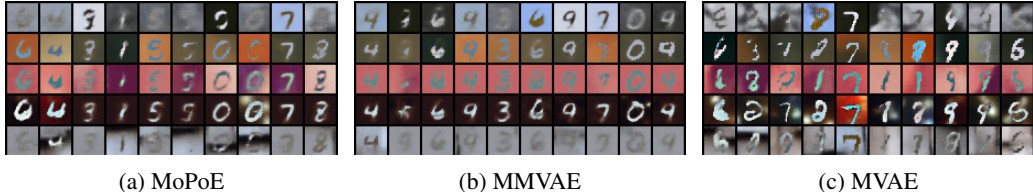

(a) MoPoE           (b) MMVAE           (c) MVAE

Figure 9: Ten unconditionally generated images from the respective five modalities for each model. Column-wise, we use the same latent codes, sampled from the prior. Note that, row-wise, the digits should not be ordered.

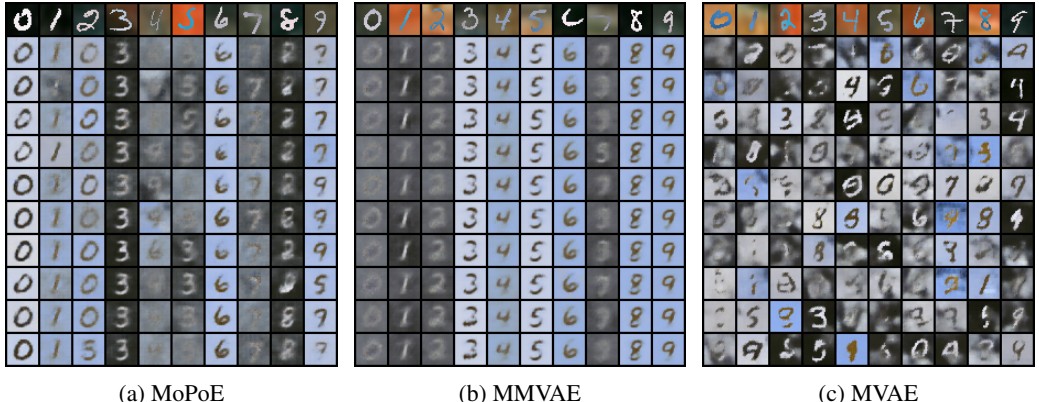

(a) MoPoE           (b) MMVAE           (c) MVAE

Figure 10: Conditionally generated images of the first modality given the respective test example from the second modality shown in the first row. Column-wise, we take different samples from the approximate posterior, which should result in stylistic variations for generated outputs, but which should ideally not change the digit labels.

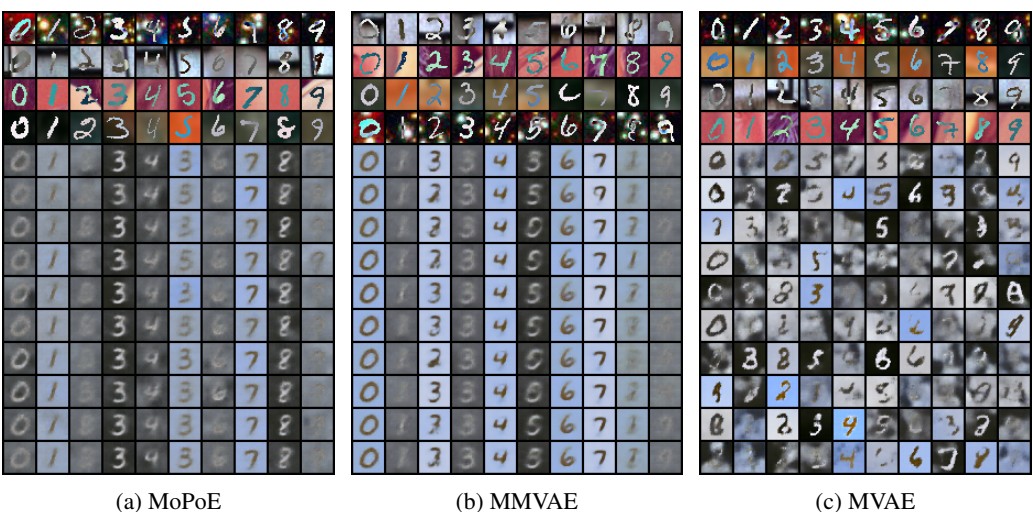

(a) MoPoE           (b) MMVAE           (c) MVAE

Figure 11: Conditionally generated images of the first modality given the four test examples from the remaining modalities shown in the first four rows. Column-wise, we take different samples from the approximate posterior, which should result in stylistic variations for generated outputs, but which should ideally not change the digit labels. Compared to the results from Figure 10, the MoPoE-VAE generates more coherent samples when conditioned on four instead of one input modality.

# E  BIMODAL CELEBA

## E.1  DATASET

The bimodal version of CelebA was introduced by Sutter et al. (2020). The text modality consists of strings which concatenate the attributes which are present in a face. If an attribute is not present, it is not present in the string which makes it a more difficult modality. Example strings can be seen in the top of Figure 4.

### E.1.1  MODALITY-SPECIFIC LATENT SPACES

Modality-specific spaces empirically have empirically shown to be useful (Bouchacourt et al., 2018; Hsu & Glass, 2018; Daunhawer et al., 2020; Sutter et al., 2020)—especially for the generative quality of samples. As CelebA is a visually challenging dataset, we adopt this idea and the ELBO formulation changes accordingly. For details, we refer the reader to the beforehand mentioned papers. The latent space is divided into a shared space $q_{\phi_c}(\boldsymbol{c}|\mathbb{X})$ and modality-specific spaces $q_{\phi_{s_j}}(\boldsymbol{s}_j|\boldsymbol{x}_j)$ for every modality $\boldsymbol{x}_j$. This allows every modality to encode information—which is specific to this modality—in a separate latent space.

$$\widetilde{\mathcal{L}}(\theta, \phi; \mathbb{X}) = \sum_{j=1}^{M} E_{q_{\phi_c}(\boldsymbol{c}|\mathbb{X})}[E_{q_{\phi_{s_j}}(\boldsymbol{s}_j|\boldsymbol{x}_j)}[\log p_\theta(\boldsymbol{x}_j|\boldsymbol{s}_j, \boldsymbol{c})]] \tag{34}$$

$$- \sum_{j=1}^{M} D_{KL}(q_{\phi_{s_j}}(\boldsymbol{s}_j|\boldsymbol{x}_j)||p_\theta(\boldsymbol{s}_j)) - D_{\mathrm{KL}}(\frac{1}{2M}\sum_{\mathbb{X}_k \in \mathbb{X}} \tilde{q}_{\phi_c}(\boldsymbol{c}|\mathbb{X}_k)||p_\theta(\boldsymbol{c}))$$

where $q_{\phi_c}(\boldsymbol{c}|\mathbb{X}) = \frac{1}{2M}\sum_{\mathbb{X}_k \in \mathbb{X}} \tilde{q}_{\phi_c}(\boldsymbol{c}|\mathbb{X}_k)$ models the shared information and $q_{\phi_{s_j}}(\boldsymbol{s}_j|\boldsymbol{x}_j)$ the modality-specific information for every modality.

All posterior approximations, shared and modality-specific, are again assumed to be Gaussian distributed, see Appendix B.

## E.2  EXPERIMENTAL SETUP

The latent spaces are set to 32 dimensions for the shared space as well as the modality-specific spaces, resulting in 64 dimensions per modality in total. We set $\beta = 2.5$ for all runs and models. All models are trained for 200 epochs. Again we use the same architectures as in Sutter et al. (2020): the encoders and decoders of both image and text use residual blocks (He et al., 2016). We use an Adam optimizer (Kingma & Ba, 2014) with an initial learning rate 0.0005. The architectures can also be looked up in the released code. The classification of samples and representations are evaluated using average precision due to the imbalanced nature of the distribution of labels.

## E.3  ADDITIONAL RESULTS

We show the attribute-specific evaluations in figs. 12 and 13 where the representations and generated samples are evaluated specific to individual attributes. The evaluations are performed for all subsets of modalities. We see the differences in averagea precision between attributes in the coherence of samples as well as the latent representations. The correlation between learned representation and coherence of samples gives further evidence on the importance of a good representation—also for the multimodal setting and its task of conditional generation.

Figure 14 displays qualitative results of randomly generated samples. We can see the high quality samples the proposed model is able to generate which cover a wide variety of attributes. In the images, minor artefacts can be seen. This suggests that there is still room for improvement doing a more rigorous hyper-parameter search.

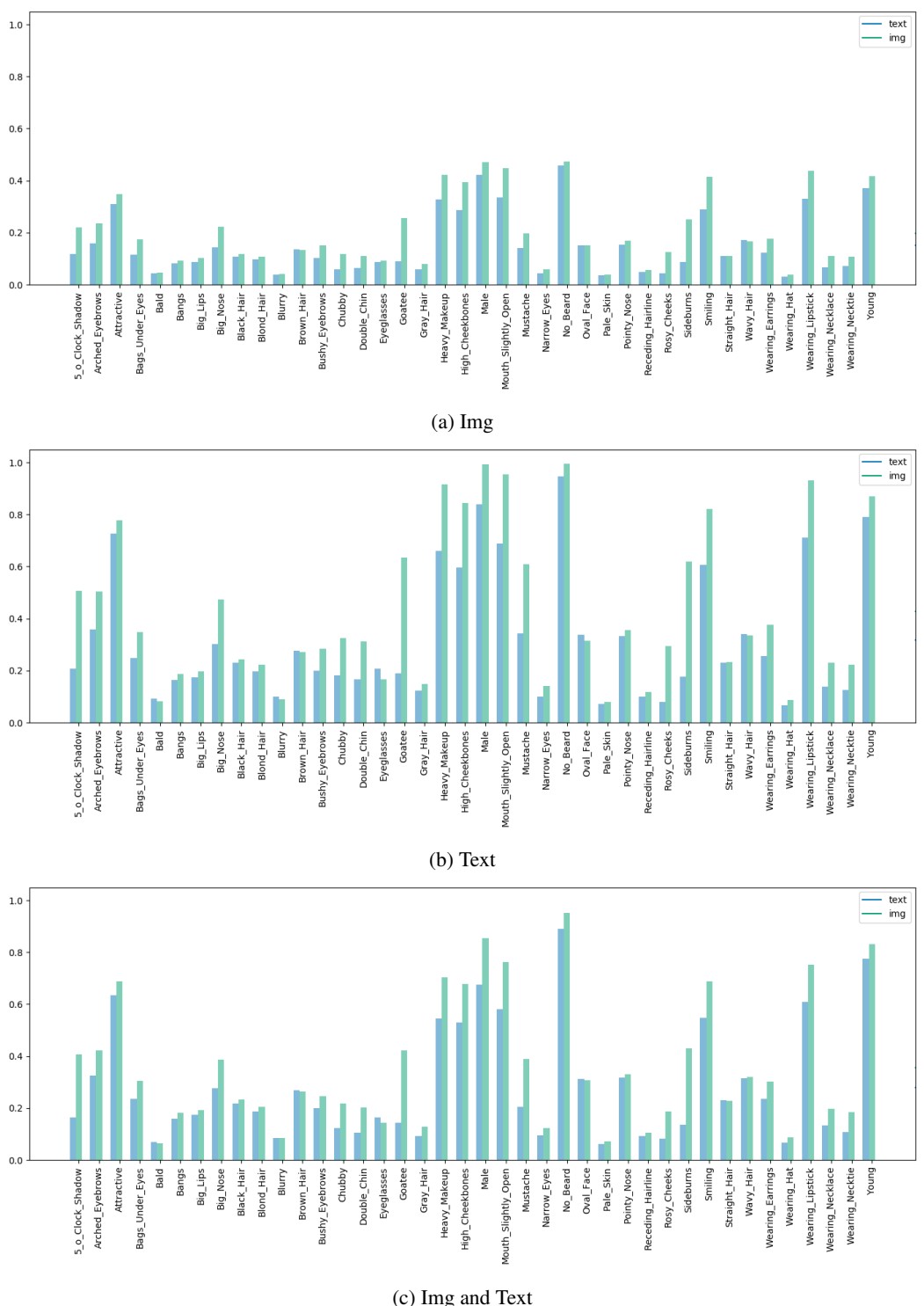

(a) Img

(b) Text

(c) Img and Text

Figure 12: Coherence of generated bimodal CelebA samples. For every subplot, image and text are generated conditionally by the the modality or subset of modalities in the caption. We see that different attributes are not learned equally well.

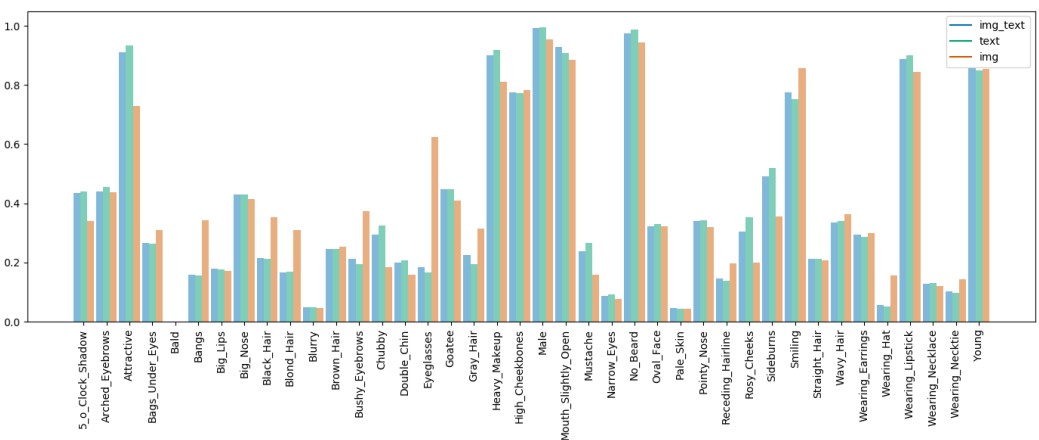

Figure 13: Learned Latent Representations for the bimodal CelebA dataset.

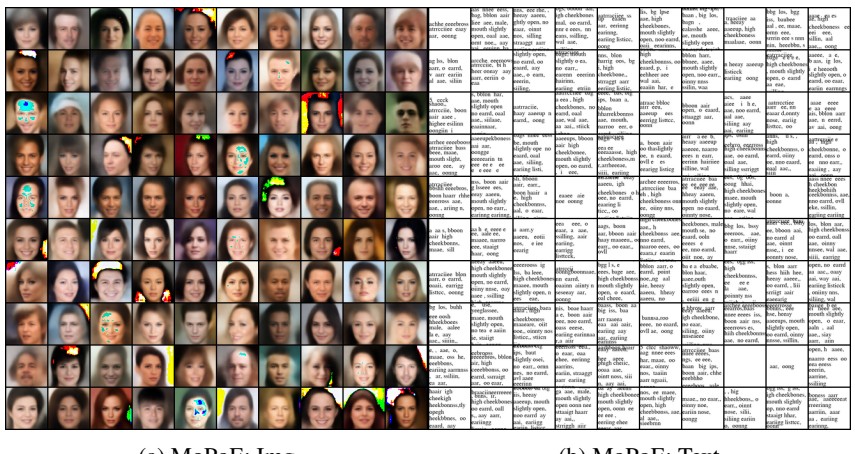

(a) MoPoE: Img            (b) MoPoE: Text

Figure 14: Qualitative Results of randomly generated CelebA samples.

