# OpenReview forum: "Generalized Multimodal ELBO"
_ICLR.cc/2021/Conference — ICLR 2021 Poster_

### Official Review · AnonReviewer1 · 2020-10-27
**Generalizes two previous works, thorough analysis, but incremental improvement**

**Rating:** 7
**Confidence:** 4

**Review:**

**Summary**: The paper combines ideas from two previous works (MVAE and MMVAE) to propose a new multimodal formulation of the ELBO for VAEs. The approximate posterior consists of a mixture of subsets, with each subset a product of approximate posteriors for each modality. This generalizes previous works, and the authors show that their objective yields a lower bound on the full data log-likelihood. They also compare their approach with MVAE and MMVAE on three multimodal datasets, showing benefits in classification accuracy, coherence, and log-likelihood. Although the proposed method is not superior in all cases, it generally achieves a reasonable trade-off across performance metrics.

**Strong Points**: Overall, the paper is fairly clear in its descriptions. For instance, it is clear that the mixture-of-products-of-experts (MoPoE) approximate posterior is a generalization of those from MVAE and MMVAE. This generalization aspect is one of the main selling points of the paper, as it illuminates the implicit assumptions/trade-offs in previous works. By combining MVAE and MMVAE under one framework, this may provide insights to researchers in this area.

The experimental analysis is thorough and rigorous. The authors use two recently proposed multimodal benchmark datasets and propose their own dataset, PolyMNST. The benefits of MoPoE is quantified on three metrics: classification accuracy, coherence, and log-likelihood. These are reported using subsets of the data modalities. A variety of experimental results are also shown in the appendix along with error bars (over 5 random seeds).

The authors provide experiment details, primarily in the appendix, for reproducibility. Although some aspects are omitted (e.g. architectures, optimizers, etc.), many details are present. Likewise, the fact that the authors report performance intervals using multiple seeds is helpful for reproducibility.

The paper is perhaps not entirely novel, as it simply combines the PoE from MVAE with the MoE from MMVAE. Nevertheless, identifying these generalizations and analyzing the performance trade-offs is still somewhat novel.


**Weak Points**: Although the mathematical formulation is clear, this paper would benefit from a diagram and/or an algorithm box, which would help to further show the distinctions between the proposed approach and MVAE and MMVAE.

I found some aspects of the motivation to be unclear. For instance, the authors begin their presentation of MoPoE by stating that they want to minimize a sum of KL divergences, using the power set of data modalities for the approximate posterior. It seems that this is a result of wanting to consider the case where some data modalities are missing. However, in the formulation and experiments, if I am not mistaken, all of the modalities are present, at least for training.

The authors also claim to want to restrict their formulation to models that are “scalable,” i.e. models that do not use a separate encoder per modality subset. Yet, in the experiments, the authors only investigate up to five modalities, and this is only coming from an artificially created MNIST dataset. Although I am not familiar with this particular experimental area, this does not strike me as a regime where “non-scalable” models could not be applied. The datasets, generally, also seem far removed from real-world multimodal datasets, e.g. audio and video.

MoPoE does seem to improve the performance metrics in certain regimes, however, these improvements are, in many cases, incremental. That is, MoPoE does not enable any new capabilities over MoE and PoE. This is somewhat limiting in terms of the paper’s impact.

**Accept / Reject**: I would lean slightly toward acceptance. The authors do indeed provide a generalization of previous methods, along with a thorough experimental analysis across multiple datasets and metrics. While the method is not particularly novel, it is well connected to previous works and well-situated in the literature.

**Questions**: The formulation appears to rely on the assumption of missing modalities, however, if I understand correctly, all modalities are present during training. Would one expect larger differences in performance if modalities were missing during training? Is this a standard assumption in multimodal learning?

In Section 3.5, it is stated that MoPoE combines “the strengths of both MoE as well as PoE while circumventing their weaknesses.” Why is this the case? If this were the case, wouldn’t MoPoE perform uniformly better than MoE and PoE in all regimes?

**Additional Feedback**:

Introduction:
Unclear what the purpose is in the opening paragraph. Representation learning in general?
It would be helpful to define semantic coherence more precisely upfront.
Similarly, abstract mean function should be defined more precisely upfront.

Related Work:
At what point does a model become non-scalable. Could non-scalable models handle the datasets in this paper?

Method:
Typically, the log marginal likelihood (top of page 3) is expressed as an average, not a sum.
Should explain what $q_\phi$ is upfront. It is not mentioned until halfway through the next paragraph.
The formulation involves all subsets of the power set, however, this seems to go against the assumption that some modalities are missing.
Lemma 3: Should start with “Maximizing $\mathcal{L}$…”

Experiments & Results:
Why should we care about coherence?
Shouldn’t $\beta=1$ yield the best log-likelihood estimate.
Why is MoPoE not uniformly better (e.g. Table 2, 3, Figure 3, etc.)?
Table 4 caption should be “Classification and coherence results…”

---

### Official Review · AnonReviewer2 · 2020-10-28
**Review of Generalized Multimodal ELBO**

**Rating:** 6
**Confidence:** 3

**Review:**

This paper formulates a multimodal ELBO as a mixture of product of experts. This allows them to use one encoder per mode while still allowing inference over any subset of modes without needing a new encoder for each subset. The idea is simple and appears to improve on baselines derived from a mixture or a product of experts.

At a high level I think the paper is fine, simple to understand, and the results are straightforward and good. However, the paper maybe dedicates too much space to explain this simple idea and not enough space analyzing the ELBO, e.g., looking at the tightness of the bound in comparison to baselines.

Some remarks:
P1:
* First sentence talks about generalization, but it's not really substantiated in any of the experiments. It just seems thrown in there without a lot of support.
* I have to start with that I disagree that VAEs are self-supervised models (or at least I don't gain anything by saying they are self-supervised generative models). I would omit this from the paper as it's confusing in the context of clearly self-supervised models (e.g., ADMDIM, MoCO, SimCLR, CPC, etc) and unnecessary. I'm happy to be convinced otherwise, but it might not be worth it for this paper.
* So there's a lot of 2^M for the power set over modalities, but do you really use the null set? Maybe it should be 2^(M-1)?
* Maybe explaining what abstract mean is because it's not a common term.
P2:
* Section 2: again, self-supervised
P3:
* 3.2: again, 2^M: is this correct?
P4:
* 3.3: "sharper than any of its experts" I feel like a citation is needed here. Mind going into a little detail why this is and why this is a problem?
P5:
* "while circumventing their weaknesses": could you clarify what you mean here?

Finally, I'm curious if any analysis is possible on the lower bound, e.g., tightness compared to lower bounds used in MoE and PoE works.

---

### Official Review · AnonReviewer3 · 2020-10-29
**This paper focuses on providing a more generalize multimodal ELBO to encompass previous PoE and MoE as special cases and combines their benefits.**

**Rating:** 6
**Confidence:** 4

**Review:**

This paper focuses on providing a more generalize multimodal ELBO to encompass previous PoE and MoE as special cases and combines their benefits. To this end, the authors first define the new ELBO L_{MoPoE} which is an interesting extension of PoE and MoE. Different from PoE (product of experts) and MoE (mixture of experts), MoPoE (mixture of product of experts) explores a more general way by mixing more experts where each expert is the product of a subset of all modalities’ posterior. In this way, as illustrated by the author, PoE and MoE can be seen as specific cases of MoPoE easily. The proposed model achieves competitive results compared with PoE and MoE.

Although the proposed MoPoE is exciting, the following concerns still need to be clarified to avoid misunderstandings:

The proof of Eq. 18. Jensen's inequality can be simply written as $f(E[x])≤E[f(x)]$, where $f$ is a convex function. The equation on the left hand can be seen as calculating the mean of $f(q(z|X))$ while on the right hand the mean of $q(z|X)$ is calculated first and then a function $f$ is used on it. According to my derivation, an error occurred in the direction of the inequality sign in Eq. 18. I hope that the author can check the inequality and give a more detailed derivation to avoid confusion.

At the same time, intuitively, I think Eq. 18 may be problematic. For the two implementations represented by Eq. 19 and Eq. 21, Eq. 19 is more concise and tighter. In my opinion, it is difficult for these two advantages to happen at the same time.

Minor concerns:
1.	Using all subsets of all modals is complicated. For example, when we have $n$ modals, $M$ is equal to $n^2$. It will be better for the authors to explain this in the paper.
2.	The Sub-sampled Training Paradigm strategy used in MVAE (Wu & Goodman, 2018) has a similar effect to Formula 5. The author should discuss and analyze this.
3.	The samples generated in Fig. 14 need to be discussed.
4.	Regardless of whether Lemma 3 can be proved, a verification test may be a good choice. An experiment implemented using Eq. 5 can be added to better illustrate the effect of the model proposed by the author.
5.	Although as the author stated that MoE is a special case of MoPoE, the loss function in Eq. 9 and MMVAE model (Shi et al., 2019) (the first equation on page 5) are different. The authors can give some explanations.

----------------------------------------------------------------------
Update after rebuttal
----------------------------------------------------------------------
Thanks for the feedback.
Although there are some minor issues, my main concerns are addressed.

---

### Official Review · AnonReviewer4 · 2020-10-29
**easy to read, nice paper**

**Rating:** 6
**Confidence:** 4

**Review:**

Summary: This paper proposes a new, scalable formulation for the ELBO for multimodal data which generalizes previous work. The key idea is to optimize over all possible subsets of modalities (their powerset), even if only a subset of the modalities are observed.

Quality: The paper was good quality, as it was clearly written and easy to read. Experiments demonstrated the efficacy of the approach over existing baselines and the problem motivation as well as technical exposition was clear.

Significance: There is rather limited amount of work on generative modeling in multimodal applications, so this work addresses that gap.

Pros: This was a nice work which drew interesting connections between prior works (MVAE and MMVAE), provided explanations for each method’s strengths and weaknesses from a theoretical viewpoint, and proposed an approach which bridged the two. The writing was clear and easy to read.

Cons: It's not very clear to me how the PolyMNIST setting counts as having different modalities when the only thing that's changing is the background image. For the other datasets (e.g. MNIST-SVHN-Text), the separation is more clear.

--------------
UPDATE: Thanks for the clarification and the revisions to the PDF -- I will keep my score as is.

---

### Decision · Program_Chairs · 2021-01-07
**Final Decision**

**Decision:**

Accept (Poster)

**Comment:**

After a bit of discussion, all reviewers are for accepting the paper.

Strengths:
+ Clarity (agreed by R4, R2, R1). The paper is easy to read and follow the core contributions. R3 had a concern about the correctness of a derivation, which was resolved in the discussion.
+ The work solves a core problem of generative models over multimodal applications, building on prior work with mixture and product of expert models. As R1 notes: "By combining MVAE and MMVAE under one framework, this may provide insights to researchers in this area."
+ On the datasets studied, the details for reproducibility are transparent, and multiple metrics and uncertainty over the metric results are reported.

Weaknesses:
+ Multi-modality of the benchmarks. The experiments evaluate on MNIST-SVHN, "PolyMNIST", and CelebA. The latter two benchmarks are fairly arguable in whether they're really multimodal as R4 notes: e.g., CelebA has two "modalities" of image and attribute pairs. It seems arguable whether you even need multimodal approaches.
+ Scale of the benchmarks. Language models (especially with Transformer architectures) have been studied quite a bit over multiple modalities, and these works scale significantly better applying simple strategies. It remains to be seen empirically what the utility of multimodal latent variable models really are.